# Applications of Thermal Plasmas for the Environment

## Guido Van Oost

Department of Applied Physics, Ghent University, 9000 Ghent, Belgium; guido.vanoost@ugent.be

**Featured Application: Thermal plasma technology is an environmentally friendly process for the treatment of waste streams. Its wide temperature range makes it suitable for almost any chemical composition of waste.**

**Abstract:** Thermal processing such as incineration is most commonly used for the treatment of waste streams, whereby often-incomplete combustion of organic waste can lead to dangerous products in the exhaust gases. Thermal plasma technology with its wide temperature range is suitable to treat almost any chemical composition of wastes. It enables the efficient and environmentally friendly conversion of organic waste into energy or chemicals, as well as the pyrolysis of hazardous organic compounds. The limitations of conventional technologies and stricter environmental legislation on the processing of wastes make plasma technologies increasingly attractive. Priority is given to environmental quality at affordable costs and to the use of innovative thermochemical conversion technologies (gasification and pyrolysis) to contribute to sustainable development and circular economy in which waste is managed as a resource.

**Keywords:** thermal plasmas; waste treatment; problematic waste; thermochemical treatment; environmentally friendly; gasification; vitrification

## 1. Introduction

The increasing volumes of a very wide range of waste streams is a growing concern. Thermal processing such as incineration is most commonly used for the treatment of waste streams, whereby the often-incomplete combustion of organic waste can lead to dangerous products in the exhaust gases. The stricter legislation on the processing of waste streams and the limitations of conventional technologies makes plasma technologies increasingly attractive *for sustainable development*. In most industrialized countries, the order of priority for waste disposal methods is governed by the concept of the so-called waste pyramid "Ladder of Lansink", proposed by the Dutch politician Ad Lansink in 1979): (1) waste prevention; (2) reuse (of the waste as it is); (3) recycling (reuse of compounds without loss of quality) and downcycling (material reuse with loss of quality); (4) energy recovery (by incineration, gasification or pyrolysis); (5) incineration and/or destruction without energy recovery; and (6) landfill.

The states of matter are characterized by their average energy per particle (atom or molecule). Solid matter has the lowest, *plasma the highest*. This high energy (temperature) implies that plasma is chemically very active (fast chemical conversion or modification with relatively few particles). Nuclear fusion research in view of a new sustainable energy source is the most ambitious application; it has provided many new insights into the plasma medium and is a very important stimulus for plasma research. The present paper deals with the treatment of waste using thermal plasmas.

Non-thermal as well as thermal plasmas are used for the processing of materials and waste. Thermal (hot) plasmas are characterized by their high energy density and by the equal temperatures of the electrons and the heavy particles, i.e., thermal plasmas are in local thermodynamic equilibrium. Non-thermal plasmas (also called cold plasmas), on the other hand, are non-equilibrium ionized gases, which are characterized by lower energy

densities and by the large difference between the electron temperature and the temperature of the heavy particles; therefore, they are suited for the treatment of heat-sensitive materials and for low-temperature plasma chemistry. A detailed review comprising various types of non-thermal plasmas with the mechanisms of their generation and with their applications is given in [1].

Thermal plasmas [2] are also characterized by a high energy transfer rate, short reaction times for chemical reactions in the plasma, and a wide choice of plasma media. These properties make thermal plasmas suitable for a variety of industrial applications [3,4], such as cutting, welding, spraying, furnaces for metallurgy with DC arcs and graphite electrodes, metal smelting and refining, metal recovery and purification from metal-containing waste, treatment of low radioactive waste, and environmentally friendly treatment of waste with plasma torches (the subject of this article). The first applications of thermal plasmas were in the 1980s, and since then their number has increased significantly [3]. Basic research in the 1990s led to major advances in understanding the fundamental phenomena involved and to a renewed interest in the use of thermal plasmas in materials processing and the environmentally friendly treatment of waste streams [3]. The characteristics of thermal plasma waste treatment are outlined in Section 2. A few applications of thermal plasma waste treatment are given in Section 3. Some environmental aspects are discussed in Section 4. Conclusions and outlook are presented in Section 5.

## 2. Thermal Plasma Waste Treatment

Thermal plasmas [2] offer unique advantages such as high temperatures in the range of 5000 to 50,000 K, high energy density, high energy transfer rate and extremely low reaction times for chemical reactions. Different types of plasma sources are used in various plasma processing technologies. Arc discharges create a high-density, high-temperature region between the electrodes. If the gas flow in the electrode gap is high enough, the plasma jet extends beyond one of the electrodes, transporting the plasma energy to the reaction region. A plasma torch, also known as a plasmatron, is a device that generates a directed stream of thermal plasma from the nozzle. An overview is given in [2]. The primary source of electricity for plasma torches can be direct current (DC), mains frequency (50 Hz) alternating current (AC), radio frequency (RF) alternating current, or microwaves. Other features of plasma torches include arc stabilization mechanism, plasma gas, flow type, and electrode geometry and cooling. Thermal plasmas used in waste treatment and material gasification systems are commonly produced in DC or AC electric arc discharges operating in a non- transferred mode (both electrodes are parts of the torch, and the plasma jet exits the torch through an exit nozzle). The main problem of arc discharges is erosion of electrodes and nozzles. The lifetime of inductively coupled RF discharges and microwave discharges, which operate without electrodes, is higher, but the complexity and cost of power supplies are the main limitation for high-power systems.

Thermal plasmas (produced in plasma torches) offer an alternative and superior solution for the treatment of waste streams. Organic components are converted into a calorific syngas (a mixture of mainly hydrogen and carbon monoxide), while the inorganic components can be converted into a slag directly in the process. Due to the unique property of strongly intensifying the energy content of the process gas, plasma torches offer very clear advantages over traditional combustion which relies on the energy content of the waste as a heat source. In the plasma torch, the process heat is supplied directly by heat transfer through the electric arc discharge. The use of electrical energy also enables the allothermic mode of operation, reduces the required oxygen supply (which is normally injected as air along with a lot of nitrogen, increasing the volume of the waste gas to be treated) and allows better control of the chemical processes. Another advantage is the ability to produce a syngas free of nitrogen by providing exactly the amount of gasifier (e.g., $CO_2$) to allow for carbon volatilization or solid carbon production.

The feed rate of the (pre-treated) waste into the reactor depends on the rate of heating and decomposition of the material and chemical reaction rates in the gases produced. The

residence time of the gaseous products of the decomposition of the waste must be high enough for complete dissociation of the gas molecules produced and for the chemical reactions leading to the formation of syngas molecules. A number of chemical reactions take place after the mixing of the plasma with gases, created by volatilization of the treated material. The reaction rate is enhanced by the high temperature and the presence of radicals and ions. Proper mixing of the plasma with the reactor atmosphere and efficient heat transfer to the surface of the material to be treated are necessary. The heating of the material and its volatilization are macroscopic processes with characteristic times well above the time constants of the reactions at high temperature. Hence, the gasification rate is determined primarily by the rate of heating and volatilization of the material. The conditions in the reactor volume must result in complete mixing of all components, and the resulting temperature must be high enough to ensure the proper reaction products composition.

The amount of added oxygen (oxygen, air, water, steam or $CO_2$) determines the type of plasma-assisted thermochemical process in the reactor.

- *Pyrolysis* if no oxygen is added (plasma supplies all energy needed for destruction and volatilization of material and for chemical reactions between the produced gases and the plasma)
- *Gasification* if oxygen is added to balance the number of carbon and oxygen atoms to suppress the production of solid carbon and provide maximum production of CO and maximum heating value of the produced gas. The combination of different oxidizing agents can control the ratio of hydrogen to carbon monoxide, characterizing the syngas.
- *Combustion* if oxygen is added for the production of $CO_2$ and $H_2O$, and energy is produced through oxidation (a small part of the energy is supplied by plasma).

The treatment of raw materials is optimized with regard to the quality of the syngas and the recovery process according to criteria determined by the reaction products end use, namely the energy content of the syngas for combined electricity/heat production (*thermal conversion*), or sustainable recovery of valuable products from the syngas such as hydrogen, methanol, synthetic fuel feedstock, ammonia, olefins or other liquid hydrocarbons via Fischer–Tropsch synthesis (*chemical conversion).*

The high temperatures in the reactor volume volatilize the organic material components and dissociate molecules of the produced gas into atoms and/or simpler molecules. Gases produced by material volatilization react with plasma components, and the resulting reaction product's composition depends on the temperature in the reactor volume. The temperature is determined by the energy balance between energy carried by the plasma, energy used for material heating, energy used or produced by chemical reactions, and energy spent for melting of inorganic components.

In the Treatment of Waste, Plasma Plays the Following Roles:

- Transport of energy into the reactor volume, necessary for the material decomposition, melting and volatilization;
- Energy transfer to the waste to be treated;
- Temperature control in the reactor volume;
- Reaction products composition control by supplying the necessary substances.

## 3. Applications of Thermal Plasma Waste Treatment

In recent years the number of applications of thermal plasmas has increased significantly. Basic research in the 1990s led to major advances in understanding the fundamental phenomena involved and to a renewed interest in the use of thermal plasmas in materials processing and the environmentally friendly treatment of waste streams [3,4]. The stricter environmental legislation on the processing of waste streams and the limitations of conventional technologies make plasma technologies increasingly attractive. *Priority is given to environmental quality at affordable costs* and to the use of innovative thermochemical

conversion technologies (gasification and pyrolysis) to *contribute to sustainable development and circular economy in which waste is managed as a resource.*

In Comparison with Conventional Thermal Treatment, the Main Advantages of Thermal Plasmas Treatment of Waste Streams Are:

- Much higher temperatures. The temperature inside the reactor can be controlled by the torch power, waste feed rate, and plasma gas flow rate;
- Highly reactive environment (reactive species such as atomic oxygen and hydrogen) and reducing atmosphere in the gasification process resulting in reduced $NO_x$ emissions;
- Short residence times and high throughput due to the high energy density of the plasma and the high heat transfer efficiency;
- Lower amount of plasma gas per unit of calorific power than of the gas flow in conventional technologies. Therefore, there is less loss of the energy necessary to bring the gas to the reaction temperature, and the amount of gas diluting the syngas produced is lower;
- Deep breakdown of waste into simple compounds, greatly simplifying the cleanup of harmful impurities;
- Possibility of joint treatment of different types of waste without pre-sorting, which is particularly important for the treatment of biomedical waste and other non-sorted toxic waste;
- Because high and homogeneous temperatures can be easily maintained throughout the reactor volume, the production of higher hydrocarbons, tar and other complex molecules is significantly reduced compared to combustion;
- Gasification at high temperature and rapid cooling of the synthesis gas prevent the formation of dioxins and furans (the most dangerous toxic substances);
- Low thermal inertia and easy feedback control. Possibility of quickly adapting the process by modifying the flow rate of the oxidant (air, steam or other plasma gas) and the power of the plasma torches. Ability to create the desired gas atmosphere. In an emergency a quick shutdown of the process is possible;
- Significant reduction in the volume of flue gases and, therefore, the load on the gas cleaning system, and less entrainment of dispersed particles;
- Smaller installations due to the high energy density of the plasma, the lower gas flow rates and the reduced flue gas volume;
- The heat source is electricity rather than the energy released during combustion and is, therefore, independent of the waste being processed. This provides rapid and flexible process control and more possibilities in process chemistry, including the capacity to generate valuable by-products. Easy temperature control in the reactor is made possible by changing the power of the plasma torches and the feed rate of waste and added gases;
- Optimal control of the composition of the final product in stable form. Possibility of obtaining a more calorific and cleaner synthesis gas from the organic part of the waste, which is not contaminated by the typical by-products of conventional gasification (in particular, tar);
- Vitrification of combustion ashes and production of vitrified slag usable as construction material;
- Recovery of value-added products (metals) from the slag.

Plasma gasification is a promising method for treating *solid and liquid combustible waste* [3,5]. Plasma methods have been used successfully in industry for decades. Reviews are presented in [3,4]. There are not many examples of plasma gasification on a commercial scale, but there are a large number of laboratory studies and pilot plants whose authors are unanimous in their opinion on the prospects of gasification, combustible waste plasma and the uniqueness of plasma technologies. Three examples of thermal plasma treatment of waste (*biomass, toxic organic waste and sewage sludge*) are presented below from the personal experience of the author.

Conventional *biomass gasification* technologies are based on the reaction between limited amounts of air or oxygen and solid or liquid carbonaceous material (containing mainly chemically bonded carbon, hydrogen and oxygen). The exothermic reactions release sufficient heat from the calorific value of the biomass for the production of a primary gaseous product containing mainly CO, $H_2$, $CO_2$ and $H_2O(g)$ and a small amount of higher hydrocarbons. The major problems of conventional biomass gasification are the low heating value of the syngas and the production of tar consisting of complex molecules of hydrocarbons created during the process at low temperature. In thermal plasma gasification (or plasma pyrolysis), the concentration of tar compounds in the produced syngas is much lower if the process is carried out such that the produced gas leaves the reaction zone at high temperatures (>1000 °C). Compared to conventional gasification routes (such as fluidized bed gasification), the synthesis gas produced contains only traces of tar (tens of $mg/m^3$) [5] compared to tens of $g/m^3$ with fluidized bed gasification, which enables the cleaning of syngas at temperatures of 100–200 °C without the risk of tar condensation on the filters. Biomass plasma gasification can also act as energy storage whereby electrical energy is transformed into plasma enthalpy and then stored in the produced syngas.

Thermal processing such as incineration is most commonly used for the treatment of *hazardous organic waste* (e.g., pesticides) [6] with medium to high calorific value and low halogen content. However, the often-incomplete chemical combustion of organic waste can lead to dangerous products in the exhaust gases. Indeed, the process of neutralizing organic waste by thermal methods is carried out at temperatures conducive to the formation of other harmful compounds. The use of arc plasma with average temperatures of the order of 5000 K enables the decomposition of complex organic compounds into atoms and ions at very high speeds and high conversion rate. In addition, this process can take place in the absence of oxygen, offering the possibility of carrying out plasma pyrolysis, which can have advantages in comparison with combustion.

Municipal *sewage sludge* is rich in both organic matter and phosphorus. On the other hand, it contains organic pollutants and heavy metals. From 2020 the microbiological criteria for the application of sludge in the soil are tightened, and it is likely that landfilling will be banned completely in the EU from 2024 [7]. Phosphorus is on the list of critical EU raw materials as an important nutrient, and, in some European countries, it is already legally enforceable that phosphorus must be extracted from sewage sludge. At present, however, this acquisition is hampered by the economic barrier created by the high cost of recovered phosphorus, which cannot compete with the price of the primary raw material. However, the latter is gradually increasing, which can be seen as an opportunity to store the ash from the combustion of sludge for later phosphorus recovery.

## 4. Environmental Aspects

The development of industrial thermal plasma applications has been hindered by the high energy consumption of plasma torches. Nowadays, when compared to non-plasma methods, the overall assessment of the technical and economic feasibility of thermal plasma technology must take into account its significant environmental benefits and the stricter environmental legislation. As an example, from an environmental point of view an attractive possibility is the use of high-temperature plasma technology for processing the above mentioned *problematic agricultural organic waste such as sludge* and for the valorization of the produced syngas and slag/ash to maximize the *extraction of phosphorus (see graphical abstract)* and a high control of harmful components to comply with the philosophy of the *circular economy*.

Publicly available data on commercial installations are scarce. In an interesting study [8], the impact of different processing routes for municipal solid waste (MSW) was assessed based on the environmental footprint and a multi-criteria analysis, using many environmental and sustainability indicators. In another study [9], the environmental impacts of three different advanced MSW processing pathways (gasification followed by

plasma gas cleaning, rapid pyrolysis followed by incineration, and gasification with syngas combustion) with traditional technologies are compared.

## 5. Conclusions

It can be concluded that all materials can be decomposed with thermal plasmas. The electrical energy from the torches goes to the plasma which transfers its energy to the waste in the plasma-chemical reactor, activating two simultaneous processes: the organic waste is thermally decomposed into a synthesis gas (syngas) consisting mainly of carbon monoxide and hydrogen, with a more complete conversion of C to gas than with conventional technologies), and inorganic materials are converted into an inert, non-leachable vitrified slag. Thermal plasma treatment can thus be considered as a completely closed processing system. Plasma treatment of organic waste mainly aims at the production of syngas, while the main purpose of combustion is the decomposition of materials. Thermal plasma technology is an environmentally friendly process for the treatment of waste and a very flexible tool as it can work in a wide temperature range with almost any chemical composition of waste The main advantages of thermal plasma-assisted waste processing processes over other processes are lower exhaust, greater volume reduction, smaller plant footprints, faster start-up and shutdown times, and lower costs for a given throughput.

The market of thermal plasma treatment of wastes is gaining momentum globally. The scaling-up of existing plants should be explored employing advanced plasma-based technologies Because plasma torches need a lot of electrical power, research on ways to increase the efficiency of the thermal plasma treatment processes is very important in a collective effort by scientists and engineers, environmentalists and governmental policy makers The choice of a specific thermal plasma technology depends on the type of waste, local regulations and existing auxiliary installations. Thorough computational fluid dynamics (CFD) modeling combined with chemical reaction modeling can greatly support the reactor design [5].

**Funding:** This research received no external funding.

**Institutional Review Board Statement:** Not applicable.

**Informed Consent Statement:** Not applicable.

**Conflicts of Interest:** The author declares no conflict of interest.

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
