# Peer review of "Applications of Thermal Plasmas for the Environment"

_applsci, doi:10.3390/app12147185_

Round 1

Reviewer 1 Report

The review gives some important information about the applications of thermal plasmas for the environment however as a review the author should include many and recent other researchers’ work but the number of reference and the information, tables and figures can’t be fulfilling the need of the reader to cover this interesting topic so I do not think I got a full information about the whole topic as it is considered as a review.

Author Response

References  to relevant review articles have been added. The author is one of the main authors of reference 5

Reviewer 2 Report

This manuscript provide an overview of the applications of thermal plasmas for the environment. Three important aspects have been summarized: the characteristics o thermal plasma waste treatment, a few applications of thermal waste treatment and some environmental aspects. My comments are as follows:

1.       More reference could be added for the introduction section

L57: such as cutting, welding, spraying, furnaces for metallurgy with DC arcs and graphite electrodes, metal smelting and refining, and environmentally friendly treatment of waste with plasma torches.  (give some references)

L60: The first applications of thermal plasmas were in the 1980s (provide the source or reference).

2.       In section 2, I would better to show some examples of thermal plasma devices which are already published by the researchers or used in industry.

3.       Plasmas are generated via many routes such as by high intensity arcs, by microwaves, by shock waves, by a radio frequency induction (RF) and by laser or high energy particle beams. Could the author analyze their characteristics and advantages?

4.       Only three examples of waste (biomass, toxic organic waste and sewage sludge) were presented with references 4, 5 and 6. However, to my acknowledge, nearly a thousand articles have been published for related domain. It is recommended to present much more literature study.

5.       The section conclusions and outlook is necessary for this review.

Some minor comment:

L98 CO2, the number 2 should be subscript.

L99 remove one extra colon.

Author Response

  1. More reference could be added for the introduction section L57: such as cutting, welding, spraying, furnaces for metallurgy with DC arcs and graphite electrodes, metal smelting and refining, and environmentally friendly treatment of waste with plasma torches. (give some references) L60: The first applications of thermal plasmas were in the 1980s (provide the source or reference). 2. In section

References have been added

  1. I would better to show some examples of thermal plasma devices which are already published by the researchers or used in industry.

References  to relevant review articles have been added. The author is one of the main authors of reference 5

  1. Plasmas are generated via many routes such as by high intensity arcs, by microwaves, by shock waves, by a radio frequency induction (RF) and by laser or high energy particle beams. Could the author analyze their characteristics and advantages?

Some basic information and relevant references have been added.

  1. Only three examples of waste (biomass, toxic organic waste and sewage sludge) were presented with references 4, 5 and 6. However, to my acknowledge, nearly a thousand articles have been published for related domain. It is recommended to present much more literature study.

References  to relevant review articles have been added.

  1. The section conclusions and outlook is necessary for this review.

This section has been added

Some minor comment:

L98 CO2, the number 2 should be subscript.

Done

L99 remove one extra colon

Done

Reviewer 3 Report

The submitted review exhibits Thermal processing such as incineration is most commonly used for the treatment of waste streams, whereby the often incomplete combustion of organic waste can lead to dangerous products in the exhaust gases. An alternative waste treatment process is based on thermal plasma technology, whose wide temperature range makes it suitable for almost any chemical composition of wastes and chemicals needed for the treatment of this waste, enabling the conversion of organic waste into energy or chemicals, as well as the destruction of toxic organic compounds in specific scenarios that can be considered optimal, both in terms of energy efficiency and environmental safety. The stricter legislation on the processing of waste streams and the limitations of conventional technologies make plasma technologies increasingly attractive. Priority is given to environmental quality at affordable costs, and to the use of innovative thermochemical conversion technologies (gasification and pyrolysis) to contribute to sustainable development and a circular economy in which waste is managed as a resource.

This review is well arranged, with a sequence of clear ideas, and concise writing that fits the research plan. The literature review is good and they were able to successfully discuss a discussion of their progress, from both a perspective and an applied perspective. The method they choose makes this data analysis excellent research and enables them to answer research questions and test their hypotheses. Thus, I strongly recommend this manuscript for the publication in Journal of Applied Sciences.

Author Response

Thank you very much

Reviewer 4 Report

Review of the manuscript entitled “Applications of thermal plasmas for the environment” by Van Oost, submitted for publication to Applied Sciences. The paper is an interesting review paper addressing an important environmental issue. The paper is well-written, however, not enough references was provided where technical information about plasma devices were discussed. I recommend minor revision of the manuscript before being considered for publication.

I strongly advise the author to add suitable references to previous works, given that this is a review article and more reference must be given. In addition, some minor comment are:

The last part of the featured application statement “Its wide temperature range makes it suitable for almost any chemical composition of waste and chemicals needed to treat this waste.” is unclear, please re-write and re-phrase. 

Line 10, “whereby the often incomplete”, please consider removing “the”.  

Line 12, “whose wide temperature range”, please replace “whose” with something more appropriate. 

Line 29, same issue as shown in comment #2.  

Lines 34-39, please provide appropriate references to support this statement. 

Line 41, “in particular in the fields”, please consider changing it to something like “particularly in the fields”.  

Line 41, author pointed out “nuclear fusion”, it needs to be pointed out that plasma in fusion devices would be used as a confinement layer to hold the high temperature fusion reactions for very short time. Please consider clarifying this or explain the meaning of mentioning “nuclear fusion” here. 

Lines 44-48, number of plasma applications was discussed. Please provide appropriate references to acknowledge each of these applications.

Lines 49-54, please cite number of references for non-thermal (cold plasmas), in addition it needs to be noted that the application of cold plasma are not limited to those mentioned by the author. Tooth bleaching, treatment of cancer cells, increasing polymeric joint adhesion, leukemia, and etc. are some of the important applications of cold plasmas, consider adding these to your research with appropriate references. 

In the introduction section, I highly recommend not to use very short paragraphs. Implementing changes as discussed in comment #8 and #9 could solve this issue as well. 

Line 69, “plasma units”, what do you mean by “units”, devices? Please consider changing.

First paragraph of section 2, please at least one appropriate reference for every plasma device that was discussed. It is important to have appropriate reference for this technical information about devices you have discussed.

---End of my comments---

Author Response

Review of the manuscript entitled “Applications of thermal plasmas for the environment” by Van Oost, submitted for publication to Applied Sciences. The paper is an interesting review paper addressing an important environmental issue. The paper is well-written, however, not enough references was provided where technical information about plasma devices were discussed. I recommend minor revision of the manuscript before being considered for publication.

I strongly advise the author to add suitable references to previous works, given that this is a review article and more reference must be given.

References  to relevant review articles have been added. The author is one of the main authors of reference 5

 In addition, some minor comment are:

The last part of the featured application statement “Its wide temperature range makes it suitable for almost any chemical composition of waste and chemicals needed to treat this waste.” is unclear, please re-write and re-phrase.

Done

Line 10, “whereby the often incomplete”, please consider removing “the”.

Done

Line 12, “whose wide temperature range”, please replace “whose” with something more appropriate.

Done

Line 29, same issue as shown in comment #2.

Done

 Lines 34-39, please provide appropriate references to support this statement.

Done

 Line 41, “in particular in the fields”, please consider changing it to something like “particularly in the fields”.

Done

Line 41, author pointed out “nuclear fusion”, it needs to be pointed out that plasma in fusion devices would be used as a confinement layer to hold the high temperature fusion reactions for very short time. Please consider clarifying this or explain the meaning of mentioning “nuclear fusion” here.

Introduction rewritten

Lines 44-48, number of plasma applications was discussed. Please provide appropriate references to acknowledge each of these applications.

Done

 Lines 49-54, please cite number of references for non-thermal (cold plasmas), in addition it needs to be noted that the application of cold plasma are not limited to those mentioned by the author. Tooth bleaching, treatment of cancer cells, increasing polymeric joint adhesion, leukemia, and etc. are some of the important applications of cold plasmas, consider adding these to your research with appropriate references.

Reference to review paper added

 In the introduction section, I highly recommend not to use very short paragraphs. Implementing changes as discussed in comment #8 and #9 could solve this issue as well.

Introduction rewritten

Line 69, “plasma units”, what do you mean by “units”, devices? Please consider changing.

Plasma sources

First paragraph of section 2, please at least one appropriate reference for every plasma device that was discussed. It is important to have appropriate reference for this technical information about devices you have discussed.

Appropriate references have been added

Round 2

Reviewer 1 Report

Dear Sir,

the author address my suggestions although some extension related to the comparison of recent work could be useful for the readers.

Regards